# Monitoring Maize Leaf Spot Disease Using Multi-Source UAV Imagery

Xiao Jia [1,2,3,†], Dameng Yin [2,3,†], Yali Bai [2,3], Xun Yu [2,3], Yang Song [2,3], Minghan Cheng [2,3], Shuaibing Liu [2,3], Yi Bai [2], Lin Meng [2,3], Yadong Liu [2,3], Qian Liu [2], Fei Nan [2,3], Chenwei Nie [2,3], Lei Shi [2,3], Ping Dong [1], Wei Guo [1,*] and Xiuliang Jin [2,3,*]

1 College of Information and Management Science, Henan Agricultural University, Zhengzhou 450046, China; peng6688@stu.henau.edu.cn (X.J.); dongping@henau.edu.cn (P.D.)
2 Institute of Crop Sciences, Chinese Academy of Agricultural Sciences, Beijing 100081, China; yindameng@caas.cn (D.Y.); yali.bai@wur.nl (Y.B.); 82101211308@caas.cn (X.Y.); 008170@yzu.edu.cn (M.C.); liushuaibing@whu.edu.cn (S.L.); by@imde.ac.cn (Y.B.); 82101231399@caas.cn (L.M.); liuyadong@caas.cn (Y.L.); qian999.liu@polyu.edu.hk (Q.L.); z20213539@stu.sxau.edu.cn (F.N.); niechw@zjweu.edu.cn (C.N.); shilei02@caas.cn (L.S.)
3 National Nanfan Research Institute (Sanya), Chinese Academy of Agricultural Sciences, Sanya 572025, China
* Correspondence: guowei@henau.edu.cn (W.G.); jinxiuliang@caas.cn (X.J.)
† These authors contributed equally to this work.

**Abstract:** Maize leaf spot is a common disease that hampers the photosynthesis of maize by destroying the pigment structure of maize leaves, thus reducing the yield. Traditional disease monitoring is time-consuming and laborious. Therefore, a fast and effective method for maize leaf spot disease monitoring is needed to facilitate the efficient management of maize yield and safety. In this study, we adopted UAV multispectral and thermal remote sensing techniques to monitor two types of maize leaf spot diseases, i.e., southern leaf blight caused by *Bipolaris maydis* and Curvularia leaf spot caused by *Curvularia lutana*. Four state-of-the-art classifiers (back propagation neural network, random forest (RF), support vector machine, and extreme gradient boosting) were compared to establish an optimal classification model to monitor the incidence of these diseases. Recursive feature elimination (RFE) was employed to select features that are most effective in maize leaf spot disease identification in four stages (4, 12, 19, and 30 days after inoculation). The results showed that multispectral indices involving the red, red edge, and near-infrared bands were the most sensitive to maize leaf spot incidence. In addition, the two thermal features tested (i.e., canopy temperature and normalized canopy temperature) were both found to be important to identify maize leaf spot. Using features filtered with the RFE algorithm and the RF classifier, maize infected with leaf spot diseases were successfully distinguished from healthy maize after 19 days of inoculation, with precision >0.9 and recall >0.95. Nevertheless, the accuracy was much lower (precision = 0.4, recall = 0.53) when disease development was in the early stages. We anticipate that the monitoring of maize leaf spot disease at the early stages might benefit from using hyperspectral and oblique observations.

**Keywords:** multi-source imagery; UAV; maize leaf spot; random forest

## 1. Introduction

Maize (*Zea mays*) is one of the three major grain crops in the world, whose planting area and yield are lower than only wheat and rice [1]. The safety of maize production is closely related to the global food security [2]. However, the frequent occurrence of maize diseases has greatly reduced maize production [3]. Therefore, monitoring the occurrence of maize diseases and insect pests quickly and accurately is a hot topic in current research [4].

Diseases cause the destruction of crop pigments and structures, resulting in altered absorption and reflection characteristics of crops, which provides the theoretical basis and technical support for disease monitoring using remote sensing [5]. Given the advantages of

an unmanned aerial vehicle (UAV) such as fast data acquisition, flexible data acquisition time, and various sensors that can be carried, increasingly more UAV-based crop disease research has emerged [6].

Various types of data collected using UAV have been adopted to monitor crop diseases, including red-green-blue (RGB) images, multispectral (MS) images, hyperspectral (HS) images, and thermal infrared (TIR) images. High spatial resolution UAV-RGB images have been widely used for the identification and detection of crop disease spots [7,8]. In addition to the raw images, RGB data have also been transformed to color vegetation indexes, such as the normalized difference index (NDI) and green index (GI), for monitoring crop diseases [9]. UAV-MS data obtain more spectral information than RGB data, which capture the difference between healthy and diseased crops [10]. Such differences are reflected in visible light, near infrared (NIR), as well as short-wave infrared (SWIR) regions [5,11,12]. UAV-HS data have more abundant spectral information, and the captured spectral differences caused by diseases are more detailed and specific than multispectral data [13]. Therefore, UAV-HS data are generally used for early crop disease monitoring [14]. TIR imaging technology can monitor the changes in canopy temperature caused by changes in crop respiration and transpiration caused by crop diseases and insect pests. Therefore, TIR imaging technology can be used to monitor crop diseases [15].

Multi-source data collaborative disease monitoring has shown better stability and reliability than using single data sources in crop disease and stress monitoring [5,6]. Usually spectral and thermal data are combined. Typical features used include band reflectance, spectral indices, and textural indices, while more complicated features have also been adopted. The most intuitive way to combine multi-source data is to stack the original bands. For example, Sankaran et al. [16] used 13 features (including the reflectance of 12 MS bands and a TIR band) to monitor the incidence of Huanglong disease of citrus. Altogether, 36 healthy citrus trees and 38 trees with leaves that showed chlorosis and blotchy mottle were distinguished. They achieved overall accuracy of 0.87, with 0.89 specificity and 0.85 sensitivity. Compared to the original band reflectance, derived features such as spectral indices and textural indices have the potential to be more discriminative. Feng et al. [17] compared the performance of different combinations of TIR and MS features in monitoring the severity of wheat powdery mildew. The disease severity was quantified according to the fraction of disease spots to total leave area, which ranged from below 5% to over 40%. By combining nine MS spectral indices, four MS textural features, and two TIR-based indices, the coefficient of determination ($R^2$) reached 0.84, which was much higher than using the three types of features separately (0.22–0.60).

In terms of methodology, approaches that have been adopted in crop disease monitoring can be divided into two groups, i.e., deep learning and conventional machine learning. Deep learning methods are usually applied on UAV-RGB images to detect maize disease spots. For example, mask regional convolutional neural network models (mask R-CNN) have been adopted to segment and recognize maize northern leaf blight [18]. They worked on 3000 maize leaf images, which had 5234 lesions in total. The reported accuracy reached up to 0.96. However, because often each disease spot needs to be labeled, these deep learning models demand massive labeling efforts and computational resources. Therefore, such models are usually performed at the leaf scale, making them not easily applicable to large areas. Conventional machine learning methods, in contrast, are less complicated and often applied on UAV-MS and HS data to capture the disease-induced spectral changes. In their work to monitor the severity of wheat powderworm disease, Feng et al. [17] provided a comprehensive comparison of the multiple linear regression algorithm (MLR), back propagation neural network (BPNN), random forest (RF), and extreme learning machine. Through 10-fold cross validation, they found that the RF model had the highest accuracy regardless of the input features.

In contrast to the efforts into plant disease monitoring, research on UAV-based maize disease monitoring has been rare. Liang et al. [14] analyzed the spectral change of maize northern leaf blight using HS data. They collected HS images 20 days after the first

inoculation and continued until full maturity. The disease index (DI) [19] ranged from 0 to 60% in the tasseling stage, from 0 to 80% in the filling stage, and finally from 0 to 100% in the full ripeness stage. They extracted 12 sensitive bands for big leaf spot. It was found that with the increase in the disease severity, the red edge (RE) band and NIR band had a higher correlation with the disease index of maize big leaf spot. The best model accuracy for estimating DI was $R^2 = 0.84$ and root mean square error (RMSE) of 4.59. Meng et al. [20] used ground-based HS data to detect leaf-level southern corn rust disease. They developed HS indices and employed SVM to classify two incidence classes (healthy and infected) and three severity classes (light, medium, and severe). The OA for incidence and severity detection were 87% and 70%, respectively. Chivasa et al. [21], in their paper on maize streak virus disease, classified maize varieties into resistant, moderately resistant, and susceptible using UAV-MS data and RF. Maize plots with disease severity from asymptotic to very severe were all included. The OA and kappa reached 0.77 and 0.64, respectively.

Compared to the big family of plant disease monitoring, we summarize that the following investigations in UAV-based maize diseases are lacking. First, only a small number of disease types have been covered. Second, the potential of TIR data has not been fully exploited for maize. Third, there are few studies using multi-source UAV data to monitor maize leaf spot diseases at the canopy scale. Furthermore, how to combine different data effectively and efficiently in these scenarios is yet to be studied.

Therefore, we propose to investigate the canopy scale of maize leaf spot disease incidence by combining multi-source UAV data. This paper mainly explores the possibility of combining MS and TIR data in the whole process of maize disease monitoring. The main research objectives are as follows:

(1)  To explore the spectral changes of maize canopy at different stages of disease development.
(2)  To find optimal multi-source data features and classifier for identifying maize leaf spot disease incidence.
(3)  To explore the possibility of maize leaf spot monitoring at the early stage with UAV data.

## 2. Materials and Methods

### 2.1. Study Area and Field Experiment

The study area is located at the Xinxiang experimental base of the Chinese Academy of Agricultural Sciences, in Henan Province, China (35°10′ N, 113°47′ E) (Figure 1). In this region, it rains frequently in July, August, and September every year. The average annual precipitation is 573.4 mm. Meanwhile, the temperature in July and August exceeds 30 °C nearly every day, and sometimes even reaches over 40 °C. The average annual relative humidity is 68% [22,23]. Because of the high temperature and humidity, the outbreak of maize leaf spot in this region is particularly common.

A total of 246 different maize materials were planted in the experimental field, of which 221 were from breeding lines, 8 fruit maize, and 17 main varieties. We refer to each of the 246 types of maize as a material, following the protocol of breeding [24]. Each material was planted in two plots. In each plot, 24 maize plants were planted in four rows with 60 cm row spacing and 22.5 cm plant spacing. To ensure the occurrence of leaf spot diseases, we inoculated the maize in the first 246 plots with *Bipolaris maydis* and the other 246 plots with *Curvularia lutana*. *Bipolaris maydis* causes southern leaf blight [25] while *Curvularia lunata* causes Curvularia leaf spot [26], and both are widely distributed maize leaf spot diseases in China. Since our purpose was not to study the disease resistance, we did not set up a separate control group. Therefore, we had 492 sample plots.

The inoculation took place in the evening of 2 August 2021, during the big trumpet period of maize. We inoculated the maize leaves using a spray of $1 \times 10^5$ spores per mL of concentration. About 5 mL spray was applied evenly in each plot. Because the resistance of the 246 maize materials in this experiment to the two inoculated leaf spot diseases was different, healthy plots were present in each data collection.

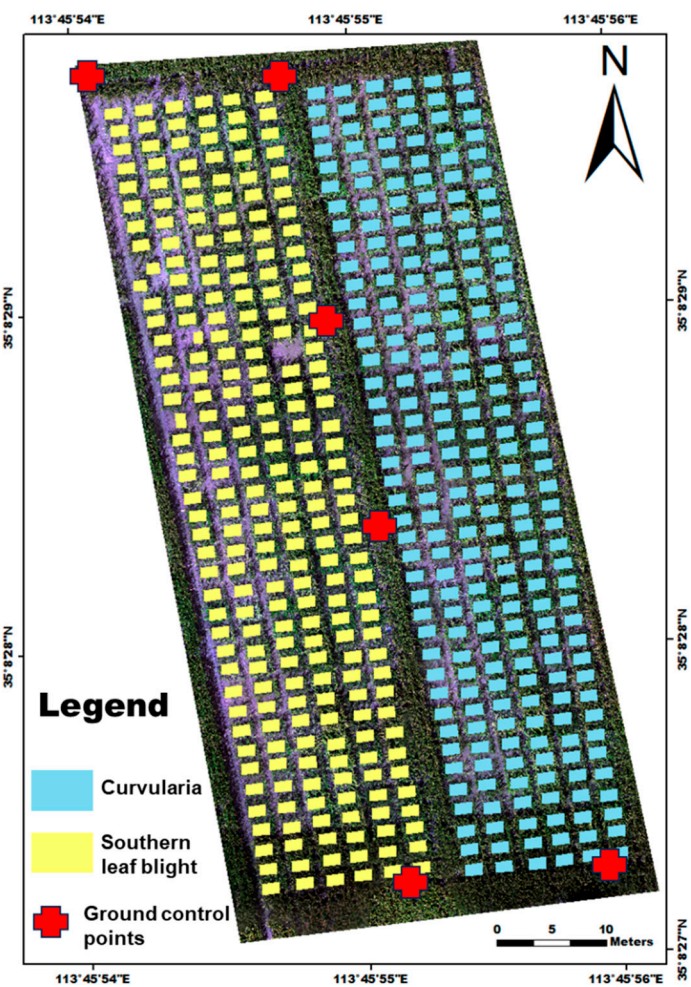

**Figure 1.** Study area.

*2.2. Data Acquisition*

2.2.1. UAV Data

DJI M600 Pro (DJI, Shenzhen, Guangdong, China), RedEdge-MX (MicaSense Inc., Seattle, WA, USA), and FLIR Duo Pro-R640 (FLIR Systems Inc., Portland, OR, USA) were used to form the UAV flight system (Figure 2).

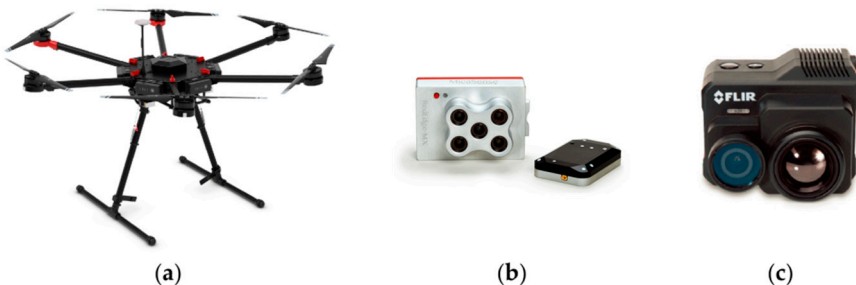

| (a) | (b) | (c) |

**Figure 2.** The UAV flight system. (**a**) DJI M600 Pro. (**b**) RedEdge-MX MS camera and the downwelling light sensor (DLS 2). (**c**) FLIR Duo Pro-R640 TIR camera.

DJI M600 Pro (Figure 2a) is a professional UAV, with a take-off weight of 9.5–15.5 kg and size of 166.8 cm × 151.8 cm × 72.7 cm. The maximum hover time is 38 min, and the maximum wind speed that can be sustained is 8 m/s.

The RedEdge-MX camera (Figure 2b) (weight: 232 g, size: 8.7 cm × 5.9 cm × 4.54 cm, resolution: 1280 × 960 pixels) used in the research is an off-the-shelf professional MS

camera with five narrow bands including blue (B), green (G), red (R), RE, and NIR. Their central wavelengths/bandwidths are 475 nm/20 nm (B), 560 nm/20 nm (G), 668 nm/10 nm (R), 717 nm/10 nm (RE), and 840 nm/40 nm (NIR), respectively. A downwelling light sensor (DLS 2) is connected to the camera to measure the ambient light and sun angle for each of the five bands of the camera, as well as to record the global positioning system (GPS) data. The ambient light and sun angle information is then used to correct for global lighting changes in the middle of a flight, such as those that can happen due to clouds covering the sun.

The FLIR Duo Pro-R640 TIR camera (Figure 2c) (weight: 1.3 kg, size: 14.3 cm × 19.5 cm × 9.5 cm, resolution: 640 × 480 pixels, FOV: 25° × 19°) used in the research is an off-the-shelf professional TIR imaging sensor with the wavelength range of 7.5–14 µm and temperature measurement range of −40 °C to +150 °C. The accuracy is +/−5 °C or +/−5% of reading within the range of −25 °C to +135 °C.

Data collections were performed four times, on August 6, August 14, August 21, and September 2, covering the early to late disease development stages (Table 1). The UAV data and field data were acquired on the same dates. The flights were taken between 10 a.m. and 3 p.m. (Table 1). Due to the simultaneous rain and heat in Xinxiang, we had to design shorter flight paths in August, resulting in different flight altitudes and spatial resolutions of the UAV data.

**Table 1.** UAV data acquisition.

| Date of Acquisition | Days after Inoculation | Disease Development Stages | MS | | TIR | |
|---|---|---|---|---|---|---|
| | | | Altitude (m) | Spatial Resolution (m) | Altitude (m) | Spatial Resolution (m) |
| 6 August 2021 | 4 | early | 70 | 0.045 | 70 | 0.106 |
| 14 August 2021 | 12 | early metaphase | 70 | 0.050 | 70 | 0.113 |
| 21 August 2021 | 19 | middle | 50 | 0.034 | 50 | 0.077 |
| 2 September 2021 | 30 | late | 20 | 0.018 | 50 | 0.076 |

### 2.2.2. Field Data

The field data included two parts, i.e., the disease survey data, and the temperature data of the environment during the flights. The temperature data were obtained from the meteorological station near the study area.

The disease survey data were collected to provide ground truth labels for the disease incidence classification. They were collected through inspecting photographs of maize leaves by pathologists, quantifying the leaf-level disease grade, calculating plot-level DI, and binarizing to two disease incidence classes (Figure 3).

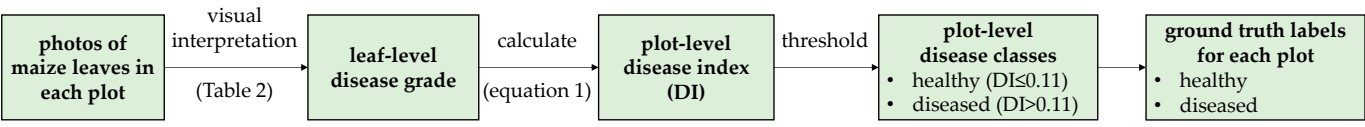

**Figure 3.** Ground truth collection through field data processing.

Due to the large workload for surveying 492 plots at each sampling date, we could not inspect each plant (24 plants per plot) but only a subset in each plot. Luckily, thanks to the spray inoculation approach, the disease development was fairly uniform within each plot. Therefore, we randomly selected two maize plants in each plot and took photos of three leaves from each plant. Before the ear shoot initiated, the top three leaves were selected. After the ear shoot appeared, the ear leaf and the leaves right above and below the ear were selected. Examples of the photos are displayed in Table 2.

**Table 2.** Disease grade of leaf spot at the leaf level.

| Disease Grade | 1 | 3 | 5 | 7 | 9 | Reference |
|---|---|---|---|---|---|---|
| Symptom description * | Disease spots account for less than or equal to 5% of the leaf area | Disease spots account for 6–10% of the leaf area | Disease spots account for 11–30% of the leaf area | Disease spots account for 31–70% of the leaf area | Disease spots cover the whole leaf and leaf dying | |
| Sample photo (southern leaf blight) | | | | | | [27] |
| Sample photo (Curvularia leaf spot) | | | | | | [28] |

* The disease grade was determined according to the proportion of disease spot area for both diseases.

According to the national standards for crop disease identification [27,28], we divided the diseased maize leaves into five grades, respectively, 1, 3, 5, 7, and 9 (Table 2). The DI was then calculated to summarize the leaf-level survey results into the plot level [19]:

$$DI = \frac{\sum (a \times b)}{n \times \sum b}, \tag{1}$$

where $a$ is disease grade, $b$ is the number of leaves corresponding to the disease grade, and $n$ is the maximum disease grade 9. In this study, we labeled the plots with DI less than or equal to 0.11 as "healthy" and the other plots "diseased" (Figure 3).

### 2.3. Leaf Spot Disease Detection Using UAV Data
#### 2.3.1. UAV Data Pre-Processing

The pre-processing of the UAV data involves the following four steps: mosaicking and calibration, co-registration, plot extraction, and calculation. Through the pre-processing, the plot-level average of bands and indices are derived. They are later used as input features for the classification models.

First, we mosaicked and calibrated the original UAV photos to generate orthomosaic images of the study area. The MS images were calibrated according to a gray board to obtain the spectral reflectance of each pixel. Similarly, the TIR images were calibrated according to a black board to obtain the temperature of each pixel. This step was performed with a commercial software, Agisoft Photoscan (version 1.4.5, Agisoft LLC, St. Petersburg, Russia).

Second, the MS and TIR orthomosaics were co-registered to eliminate spatial displacements among the images. This was implemented using the *Georeferencing* tool in ArcGIS (version 10.8, Environmental Systems Research Institute, Inc., Redlands, CA, USA) according to the ground control points (GCPs) placed in the field. The locations of these GCPs are as illustrated in Figure 1.

Next, we annotated the plot boundaries (Figure 1) in the orthomosaics and extracted the orthorectified images for each plot. This step was also implemented in ArcGIS 10.8.

Finally, we extracted and calculated the MS and TIR features of each plot (Table 3), including the band values and the commonly used indices for detecting crop diseases. The 22 MS indices were calculated directly from the reflectance of two or more MS bands to enhance the difference between healthy and diseased maize pixels. These indices have been found to be relevant to crop pigment, leaf area, and structure, and thus are highly sensitive to disease incidence. The TIR index, i.e., normalized differential canopy temperature index (NDCT), was calculated by taking the normalized difference between the canopy temperature (Tc, from the TIR band) and the air temperature (Ta, from the ground-based meteorological station). The plot average was taken for each reflectance and index to represent the features in each plot. The calculation of indices was implemented in Jupyter Notebook Python 3.7.3.

**Table 3.** The 22 MS spectral indices and one TIR index used as classification features in this study.

| Feature Type | Feature Name | Equation | Reference |
|---|---|---|---|
| MS | Anthocyanin Reflectance Index 2 (ARI$_2$) | $ARI_2 = R_{NIR} \cdot (1/R_G - 1/R_{RE})$ | [29] |
| | Chlorophyll Index (CI) | $CI = R_{NIR}/R_{RE}$ | [30] |
| | Vegetation Color Index (CIVE) | $CIVE = 0.441R_R - 0.881 R_G + 0.385R_B + 18.787$ | [31] |
| | Red Edge Chlorophyll Index (CRI$_{RE}$) | $CRI_{RE} = 1/R_G - 1/R_{RE}$ | [32] |
| | Red Chlorophyll Index (CRI$_R$) | $CRI_R = 1/R_G - 1/R_R$ | [32] |
| | Enhanced Vegetation Index (EVI) | $EVI = 2.5(R_{NIR} - R_R)/(R_{NIR} + 6R_R - 7.5R_B + 1)$ | [33] |
| | Difference Vegetation Index (DVI) | $DVI = R_N - R_G$ | [34] |
| | Greenness Index (GI) | $GI = R_G/R_B$ | [35] |
| | Green Ratio Vegetation Index (GRVI) | $GRVI = R_N/R_G$ | [36] |
| | Modified Triangular Vegetation Index 1 (MTVI$_1$) | $MTVI1 = 1.44(R_{NIR} - R_G) - 3(R_R - R_G)$ | [37] |
| | Normalized Difference Vegetation Index (NDVI) | $NDVI = (R_{NIR} - R_R)/(R_{NIR} + R_R)$ | [37] |
| | Blue NDVI (NDVI$_B$) | $NDVI_B = (R_{NIR} - R_B)/(R_{NIR} + R_B)$ | [32] |
| | Green NDVI (NDVI$_G$) | $NDVI_G = (R_{NIR} - R_G)/(R_{NIR} + R_G)$ | [32] |
| | Nonlinear Index (NLI) | $NLI = (R_{NIR}^2 - R_{RED})/(R_{NIR}^2 + R_{RED})$ | [38] |
| | Normalized Pigment Chlorophyll Index (NPCI) | $NPCI = (R_R - R_B)/(R_R + R_B)$ | [35] |
| | Optimized Soil-Adjusted Vegetation Index (OSAVI) | $OSAVI = (R_{NIR} + R_G)/(R_{NIR} + R_G + 0.16)$ | [36] |
| | Plant Pigment Radio (PPR) | $PPR = (R_G - R_B)/(R_G + R_B)$ | [39] |
| | Plant Senescence Reflectance Index (PSRI) | $PSRI = (R_R + R_B)/R_{RE}$ | [39] |
| | Renormalized Difference Vegetation Index (RDVI) | $RDVI = (R_{NIR} - R_{RED})/\sqrt{(R_{NIR} + R_{RED})}$ | [40] |
| | Structure-Intensive Pigment Index (SIPI) | $SIPI = (R_{NIR} - R_B)/(R_{NIR} - R_R)$ | [39] |
| | Simple Ratio (SR) | $SR = R_{NIR}/R_R$ | [41] |
| | Visible Atmospherically Resistant Index (VARI) | $VARI = (R_G - R_R)/(R_G + R_R - R_B)$ | [42] |
| TIR | Normalized Differential Canopy Temperature (NDCT) | $NDCT = (T_C - T_a)/(T_C + T_a)$ | [43] |

### 2.3.2. Feature Selection

For the convenience of writing, the MS bands, the TIR band (canopy temperature), and the calculated indices are collectively referred to as features. Considering that not all features contribute equally to the maize leaf spot diseases, we adopted the recursive feature elimination (RFE) [44] algorithm to select features. RFE is a machine learning feature selection algorithm, which is used to reduce the computational time and complexity of the model. It has been found to be effective for feature selection in disease monitoring [45–47]. RFE selects features in an iterative manner. In each iteration, a classification model is

built using the current optimal subset of features, and the importance of each remaining feature is evaluated. The feature with the lowest importance will be recursively removed. The iteration stops when the model accuracy converges to the maximum and the optimal feature subset is selected.

To find out an optimal set of features of maize leaf spot disease detection, we compared the following five different feature sets.

- Feature set I. All MS and TIR features mentioned in this study, including six MS bands, one TIR band, and the 23 indices in Table 3.
- Feature set II. Features selected using RFE at each of the four sampling dates separately.
- Feature set III. Features selected using RFE at all the four sampling dates, i.e., intersection of the four sets in feature set II.
- Feature set IV. All MS features, including the six MS bands and 22 MS indices.
- Feature set V. All TIR features, including canopy temperature and TIR index.

### 2.3.3. Classification

We adopted four classifiers to identify maize plots with leaf spot diseases, including the back propagation neural network (BPNN), RF, support vector machine (SVM), and extreme gradient boosting (XGBoost). They are popular classifiers that have proved to be effective in crop disease identification [5,17,19].

BPNN [48] is a multi-layer neural network trained according to the error back propagation algorithm. Through back propagation, the weights and thresholds of the network are constantly adjusted to minimize the error, which makes BPNN perform well in a nonlinear relationship analysis. The *scikit-learn 1.3.1* package was used to implement this algorithm in Python. Parameters including the hidden layer size (*hidden_layer_sizes*), the activation function (*activation*), the optimizer function (*solver*), the initial learning rate (*learning_rate_init*), and the regularization term (*alpha*) were optimized according to the validation set.

RF [49] is an ensemble learning-based classifier. It combines multiple weak decision trees, and votes to obtain the final result, so that the overall model has high accuracy and generalizability. The *scikit-learn 1.3.1* package was used to implement this algorithm in Python. Parameters including the number of decision trees (*n_estimators*), the size of the feature subsets to consider when splitting a node (*max_features*), the maximum depth of each tree (*max_depth*), the minimum number of samples for splitting a node (*min_samples_split*), and the minimum number of samples for growing a leaf (*min_samples_split*) were optimized according to the validation set.

SVM [50] is a nonlinear classifier based on kernel function, which can deal with regression, classification, and discriminant analysis problems. The *scikit-learn 1.3.1* package was used to implement this algorithm in Python. Parameters including the tolerance (0.1 to 100), kernel function (radial basis function, linear, or sigmoid), and the kernel-specific parameters were optimized according to the validation set.

XGBoost [51] is also based on decision trees like RF. Different from RF, XGBoost gradually improves the predictive ability of the model by minimizing the loss function in the training process. The *scikit-learn 1.3.1* package was used to implement this algorithm in Python. Parameters including the number of decision trees (*n_estimators*), the maximum depth of each tree (*max_depth*), the fraction of training data used in each iteration (*subsample*), the fraction of features to consider when splitting a node (*colsample_bytree*), and the learning rate (*learning_rate*) were optimized according to the validation set.

In order to ensure the stability of the model, we put 80% of the samples into a training set and the other 20% into an independent test set. The partition of the feature set was carried out using the *train_test_split* module in Python. In order to ensure the maximum accuracy of the model, 80% of the training set was used for model parameter adjustment, and the remaining training data were used for validation of model parameter adjustment. The grid search method was adopted for parameter optimization, with five-fold cross validation.

The model construction process is shown in Figure 4. First, we input feature set I (i.e., six MS bands, one TIR band, and the 23 indices in Table 3) to the four classifiers mentioned above to choose the most accurate classification algorithm. Second, the selected classifier was employed in a feature selection step to form feature set II. Third, the five feature sets were separately input to the optimal classifier to identify maize leaf spot disease incidence. The optimal feature set was then determined according to the accuracy assessment. Finally, by combining the optimal classifier and feature set, an optimal model for the maize leaf spot disease incidence monitoring was constructed.

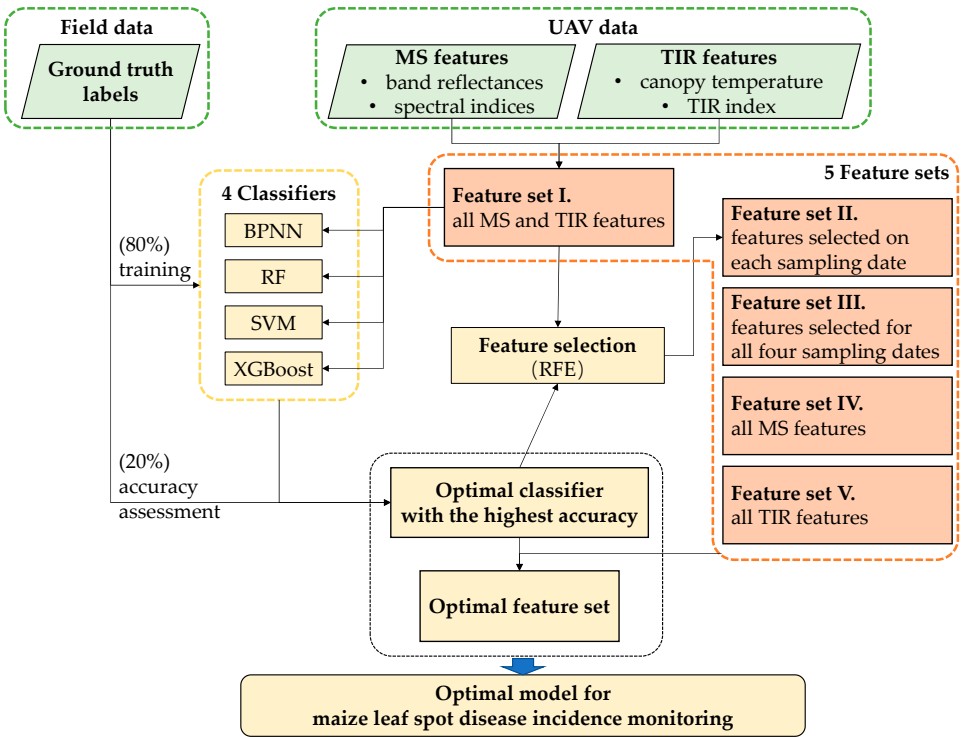

**Figure 4.** Construction of leaf spot disease incidence monitoring model.

*2.4. Accuracy Assessment*

Among the 492 sample plots, 20% was excluded from the model training and used as independent test samples for the accuracy assessment. By comparing the ground truth labels and their classified labels, we summarized these test samples into four types according to their label and correctness: True Positive (TP) is the number of correctly identified positive (diseased) samples, True Negative (TN) is the number of correctly identified negative (healthy) samples, False Positive (FP) is the number of healthy samples that are falsely identified as diseased, and False Negative (FN) is the number of diseased samples that are falsely classified as healthy [52].

We adopted four accuracy metrics to summarize the model accuracy, i.e., overall accuracy (OA) [53], precision [54], recall [54], and F1-score [55]. OA refers to the proportion of the number of samples predicted correctly with the model to the total sample. Precision is the proportion of TP in all samples that is identified as "diseased". Recall is the proportion of TP in all samples with actual "diseased" labels. F1-score is the harmonized average of recall and precision.

$$OA = (TP + TN)/(TP + TN + FP + FN) \tag{2}$$

$$Precision = TP/(TP + FP) \tag{3}$$

$$Recall = TP/(TP + FN) \tag{4}$$

$$\text{F1-score} = 2 \times \text{Precision} \times \text{Recall}/(\text{Precision} + \text{Recall}) \tag{5}$$

## 3. Results

### 3.1. Changes in the Spectra and Indices

Changes in the spectral reflectance were observed during the disease development time period in both the healthy and diseased maize (Figure 5). The reflectance of both healthy and diseased maize increased from August 6 to August 14 and decreased afterwards. Nevertheless, instead of interpreting the absolute change over time, we focus on how the difference between the healthy and diseased maize was changing.

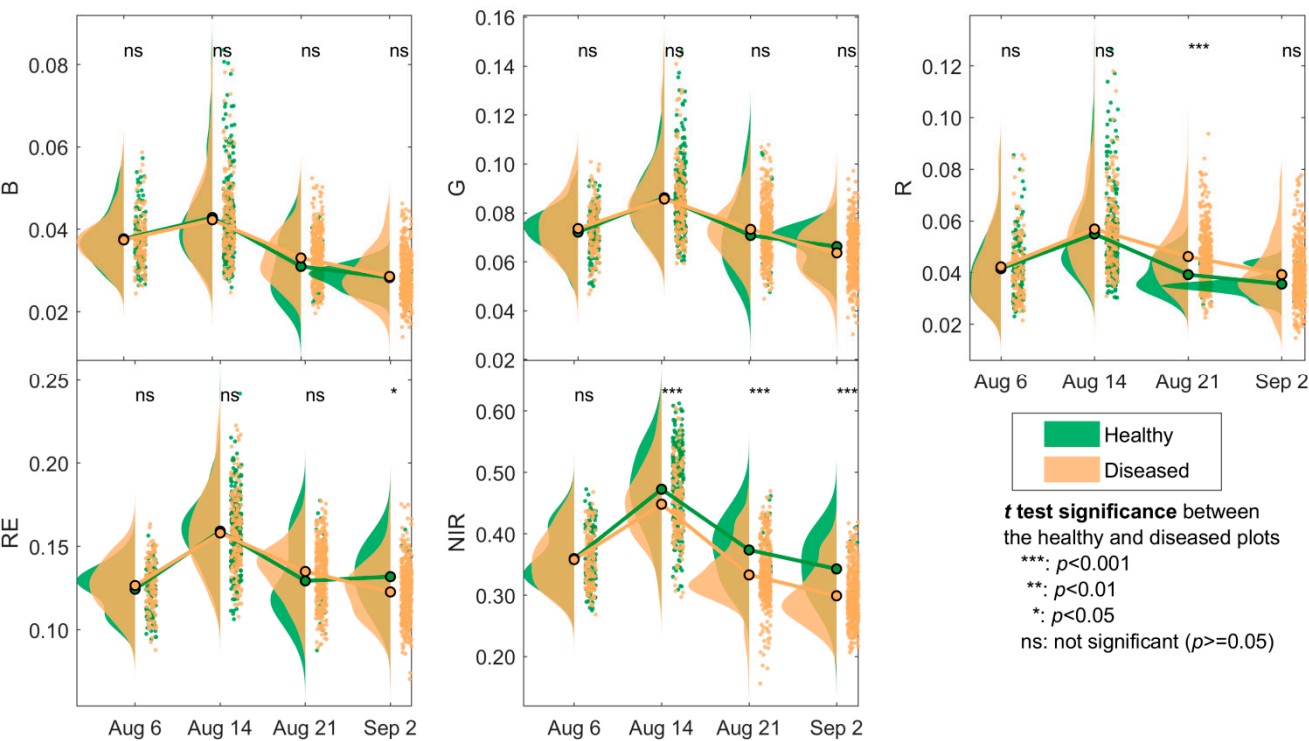

**Figure 5.** Statistical distribution of the spectral band reflectance.

Among the five bands, the largest difference between the healthy and diseased maize was observed in the NIR band. The mean reflectance of the healthy maize in the NIR band was always higher than that of the diseased maize. With the development of the disease, the difference in NIR became more and more obvious. No significant difference was observed in the B and G bands. In the R band, the mean reflectance of the healthy maize was clearly lower than that of the diseased maize. This difference became larger at the middle and late stages. In the RE band, the mean reflectance of the healthy maize was lower than that of the diseased maize in the beginning, but this pattern got reversed on September 2.

The distinction between the healthy and diseased maize was more evident in the indices than in the band reflectance (Figure 6). On August 6, the spectral characteristics were very similar in the healthy and diseased maize plots. The difference between healthy and diseased maize was not statistically significant in all the indices except PPR. The difference got more and more pronounced as the disease developed, with the largest difference on September 2.

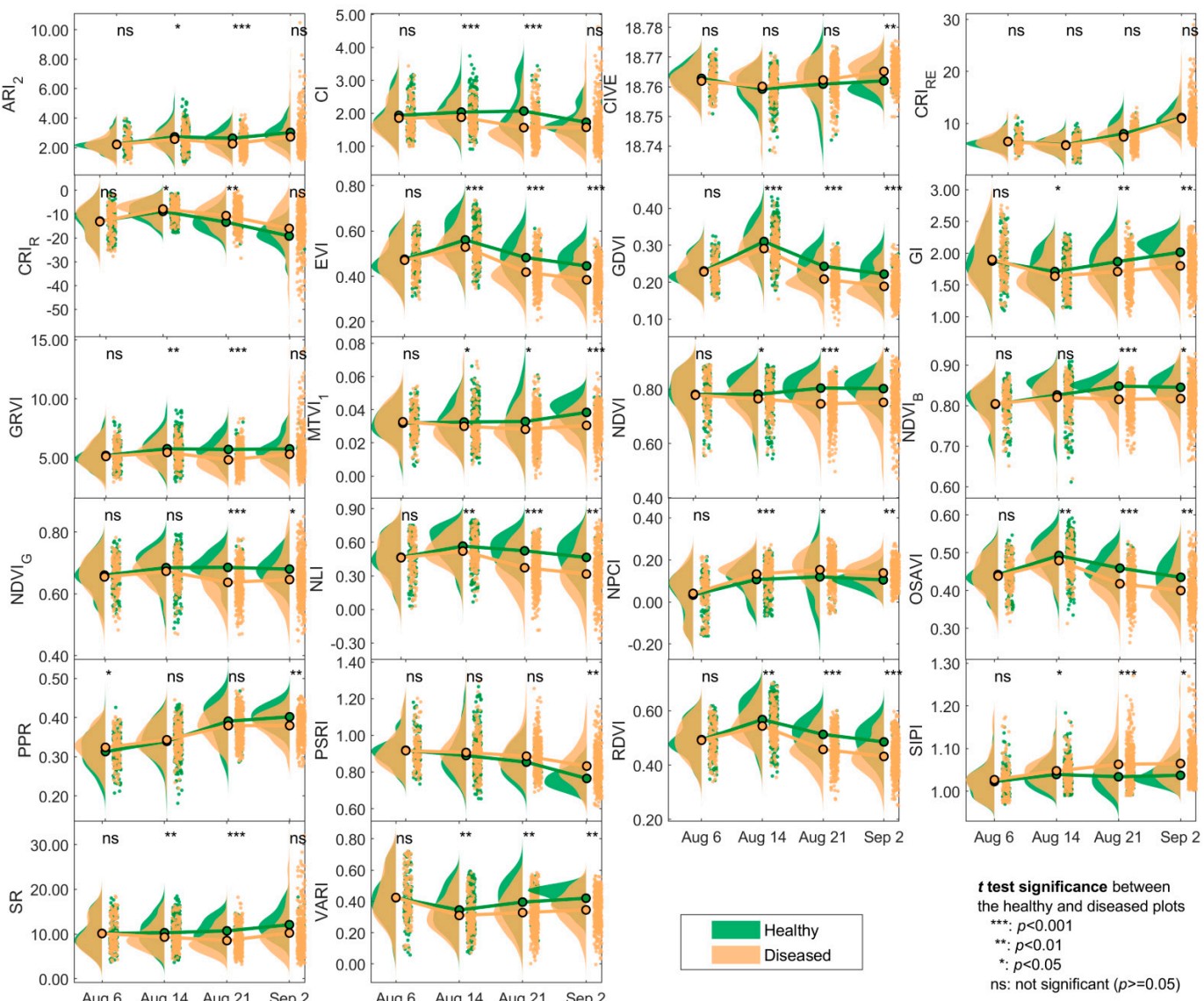

**Figure 6.** Statistical distribution of the MS indices.

Both TIR features were not found to be statistically significant in differentiating diseased and healthy maize (Figure 7). Nevertheless, the canopy temperature of the diseased maize was higher overall than that of the healthy maize. The temperature from August 6 to August 14 increased. On August 21, due to the overcast weather, the temperature is lower than August 14. On September 2, due to autumn and continuous rain from August 21 to September 2, the temperature decreased more. The temperature trend is in line with the local temperature change trend (33 °C on August 6, increased to 35 °C on August 14, and then decreased to 33 °C and cloudy on August 21 and 29 °C on September 2), indicating the validity of the TIR data.

Examining *p*-values of all the features, we noticed that the number of features that witness significant differences between the healthy and diseased maize was the largest on August 21 (20 features), followed by September 2 (18 features) and August 14 (17 features). On August 6, no features except the MS index PPR were statistically significant. Moreover, the number of features with a $p < 0.001$ significance level on August 21 was 15, which was much more than on the other sampling dates. This could probably be explained with the fact that the maize entered the milk stage afterwards, where the pigment contents in healthy maize also started decreasing. Five features were never significant. They are the reflectance

in the B band, reflectance in the G band, the MS index CRI$_{RE}$, canopy temperature, and the TIR index NDCT.

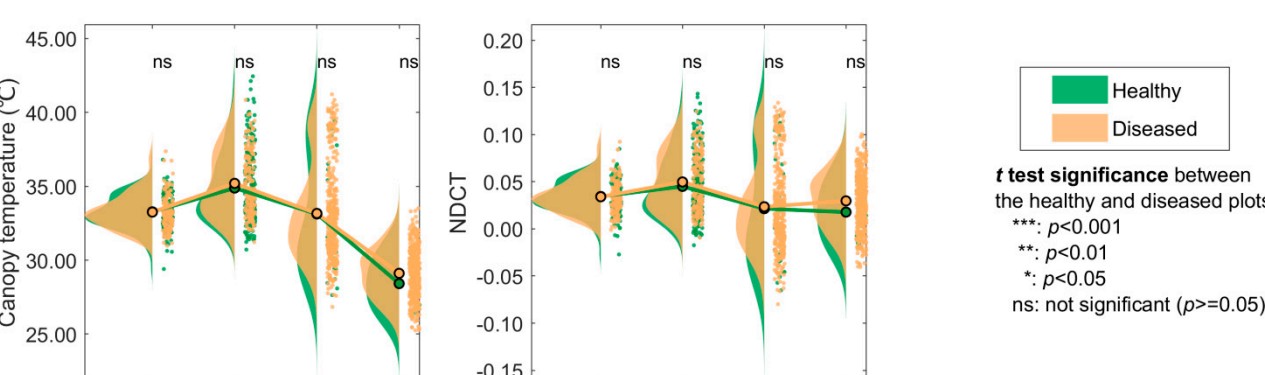

**Figure 7.** Statistical distribution of the TIR-based features.

### 3.2. Leaf Spot Disease Incidence Identification Results

Using feature set I as the input data, the performance of maize leaf spot incidence recognition models established with BPNN, RF, SVM, and XGBoost algorithms was evaluated. The results for the four sampling dates are listed in Figure 8. OA, precision, recall, and F1-score of all the classification models on August 6 were less than 0.6, while the best performing early leaf spot disease monitoring model was with RF, with an OA of 0.53. Among all the leaf spot disease monitoring models on August 14, the XGBoost model had the lowest accuracy, with only OA = 0.53 and the lowest precision (0.36) and recall (0.53). The most accurate models on August 14 were RF and SVM. SVM had the highest F1-score (0.75) while RF had the highest OA (0.62). On August 21, the performance of the four models became very close. The OA and precision were all 0.9. The recall of BPNN and XGBoost was 1, and F1-score was 0.95. The recall of RF and SVM was 0.98, and F1-score was 0.93. Among all the Sep-2 models, RF had the highest OA (0.95) with perfect precision (1.00), while the other three classifiers received a slightly higher F1-score (0.97) and recall (1.00).

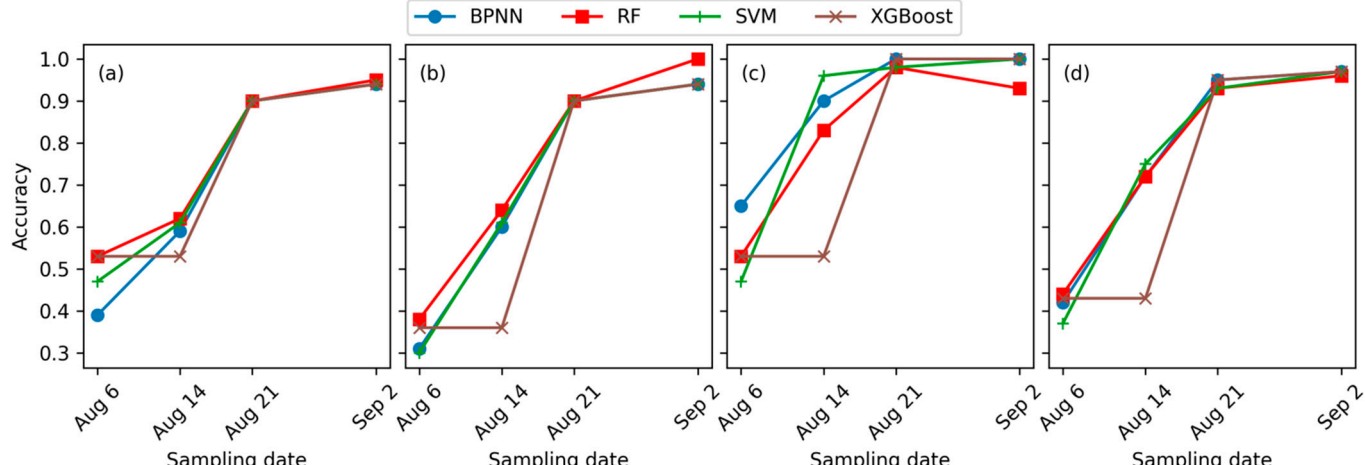

**Figure 8.** The leaf spot disease incidence identification results with feature set I as input. (**a**) OA. (**b**) Recall. (**c**) Precision. (**d**) F1-score.

Based on Figure 8, we selected RF as the optimal classifier. RF was then used for the RFE algorithm to select important features for maize leaf spot disease incidence monitoring at the four sampling dates. The feature importance results are shown in Figure 9. Features selected for each sampling date were not the same. Despite that some features did not pass the *t* test, each of the features was selected at least once in RFE. Some features were found

to be the most important for all the four sampling dates, including reflectance in three MS bands (G, RE, and NIR), 10 spectral indices (CI, CIVE, $CRI_{RE}$, $CRI_R$, DVI, GI, $MTVI_1$, NPCI, OSAVI, and PPR), and both TIR features.

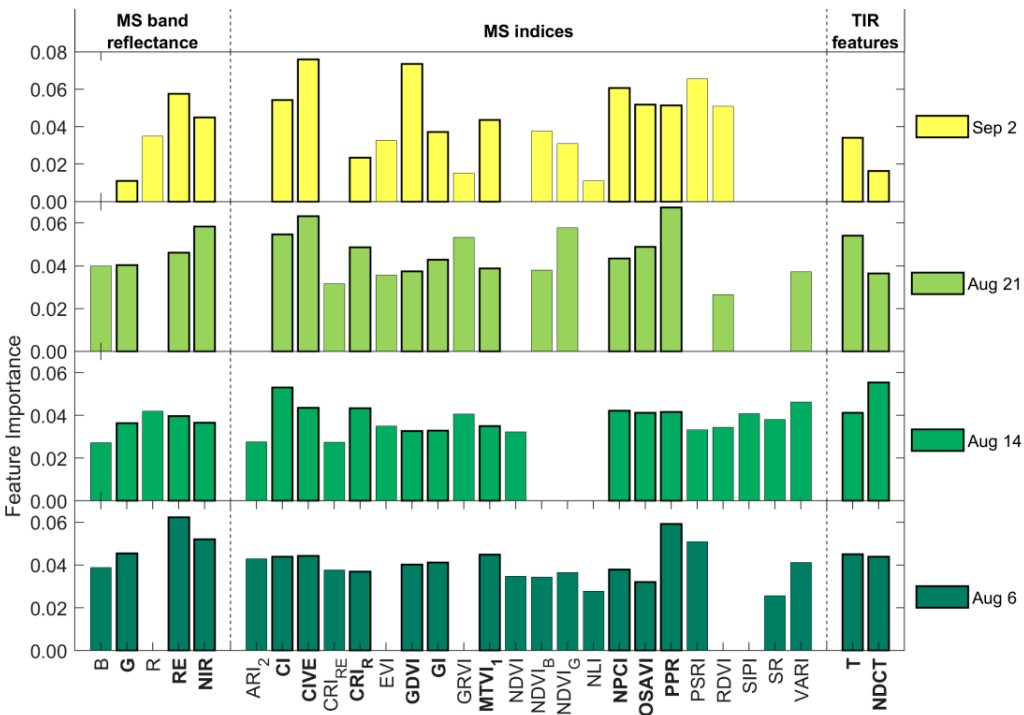

**Figure 9.** The result of feature selection with RFE using data at each sampling date. Features with no bars are those not selected for the classification. Features with names in bold fonts are those selected for all four sampling dates.

We noticed that the feature importance analysis revealed something different from the significance test. For example, the TIR features had no significant difference in the healthy and diseased maize, but were selected with RFE for all four sampling dates. The MS index PPR on August 6 has the highest feature importance, which agreed with the previous significance analysis. On the other three sampling dates, although no significant difference was found between PPR of the healthy and diseased maize, PPR had high importance in the classification model. Its importance was highest among all features on August 21.

Feature sets II and III were then formed based on the feature selection results. The five feature sets were then input into the RF algorithm and compared. Similar to the results in Figure 8, the model accuracy on August 6 and August 14 was far lower than that on August 21 and September 2 (Figure 10), indicating that the capacity of canopy-level MS and TIR observations in monitoring leaf spot was limited in the early stages of leaf spot development. This can probably be attributed to the fact that the leaf spots mainly occur in the middle and lower parts of maize in the early stage while UAV images provide nadir-view observations of the upper parts of maize. The model accuracy on September 2 was the highest regardless of the accuracy metrics used. At this stage, leaf spots developed to the upper leaves of maize, which can be easily captured in UAV images.

Comparing the performance of models using different feature sets, the model established with RF and feature set III had the highest accuracy (Figure 10). The model established with spectral features (feature set IV) was generally superior to the model with TIR features (feature set V), except on August 6. The model with the RFE selected features (feature sets II and III) had much higher accuracy than using spectral features only (feature set IV) on August 6. They became comparable at the later stages. The model with feature set V had the lowest accuracy among the five models, indicating that it is not feasible to monitor maize leaf spot disease solely based on TIR features.

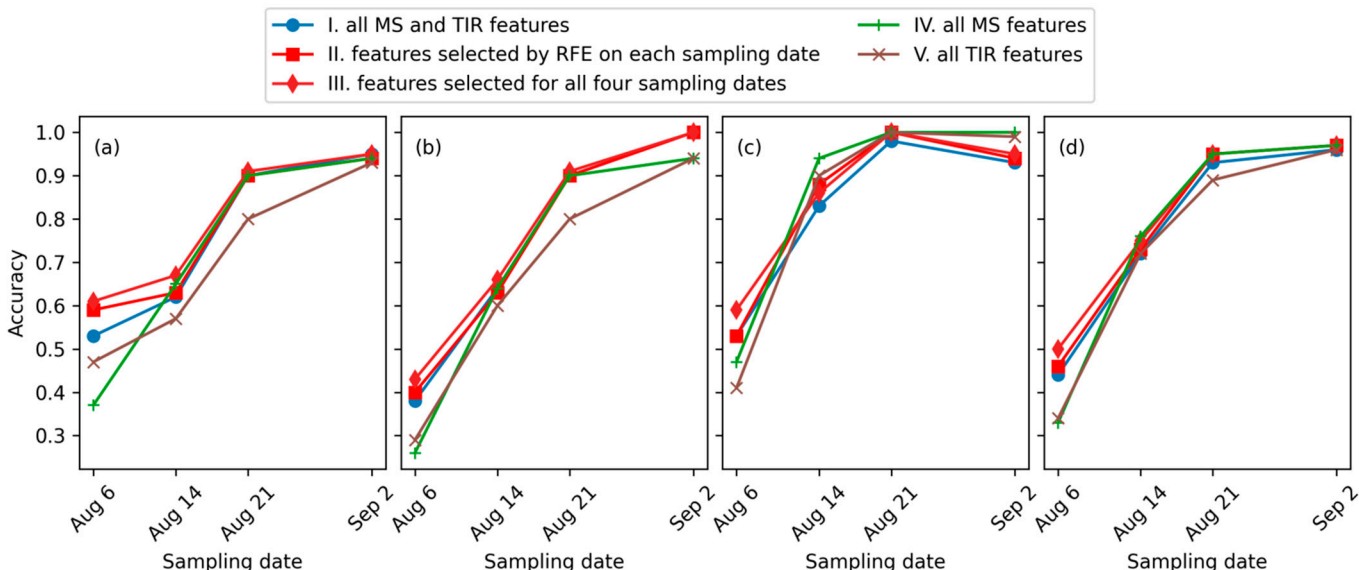

**Figure 10.** RF model accuracy using the five different feature sets. (**a**) OA. (**b**) Recall. (**c**) Precision. (**d**) F1-score.

## 4. Discussions

### 4.1. Spectral Changes of Maize Leaf Spot from Early to Late Stages

Maize leaf spot disease leads to a series of changes in the physiological, biochemical, and structural parameters in maize [20]. Typical changes include damage to the pigment in maize leaves, decreased photosynthesis, decreased leaf water content, and decreased leaf area and biomass [5]. Some of these changes coincide with the natural changes caused by tasseling and senescence [56]. Therefore, the interpretation of the spectral changes should consider the impacts of both disease incidence and natural growth.

The spectral variation trends of healthy and susceptible maize were roughly similar. The reflectance increased from August 6 to August 14 and then decreased from August 14 to September 2. Such changes are consistent with those observed by Liang et al. [14] using UAV-HS data. This could possibly be attributed to the emergence of tassels [57] as well as the change in the environment. From August 6 to August 14, maize started to enter the stage of tasseling, when the canopy coverage increased. Therefore, the spectral reflectance showed an upward trend. From August 21 to September 2, maize transited into the silking and milk stages, when the canopy coverage stopped changing. However, as the maize gradually matured and the leaves turned yellow, the spectral reflectance decreased.

As the severity of leaf spot disease increased over time, the chlorophyll damage worsened, leading to a decrease in the absorption of red light. As a result, the average spectral reflectance of healthy maize in the R band was lower than that of the diseased maize, and the spectral difference increased over time. This discovery is consistent with previous studies on wheat stripe rust, rice blast, and citrus yellow dragon disease [12,16,58]. The average spectral reflectance of diseased maize in the RE band recorded at the first three sampling dates was higher than that of healthy maize and lower than that of healthy maize on September 2. This is because the disease was so severe that the diseased maize withered due to water shortage, resulting in a "blue shift" phenomenon [14]. The spectral reflectance of August 21 and September 2 was lower than that of August 14 due to the frequent rain raising the water content in the soil and the low flight altitude, as well as the increased disease severity. The spectral differences between healthy maize and leaf spot disease maize in the B and G bands were not significant.

Analyzing the reflectance of healthy and diseased maize using UAV-based spectra can help to understand the spectral response mechanism of maize leaf spot disease, providing a basis for monitoring maize leaf spot disease and facilitating precise control of crop diseases.

This is because compared to analyzing the spectral reflectance of maize leaves under laboratory conditions, UAV-based observations in the field have the complex environmental background considered, including the influence of weather, changes in respiration and transpiration rate caused by stress on maize itself, and the influence of soil. Therefore, analyzing the reflectance of the maize canopy using UAV is closer to the actual application situation. This insight echoes the literature in detecting wheat yellow rust [59].

### 4.2. Optimal Model for Maize Leaf Spot Disease Incidence Monitoring

This study demonstrates an optimal model for efficient canopy-level maize leaf spot disease incidence monitoring, by promoting multi-source data and a machine learning algorithm. Although research on UAV-based maize leaf spot disease monitoring has been rare, this UAV multi-source remote sensing-based approach has proved effective in monitoring other diseases [5,6]. The maize diseases that have been investigated using UAV remote sensing include several types such as maize northern leaf blight [14,18], southern corn rust [20], and maize streak virus [21]. Our research provides an efficient approach for the detection of maize disease types that have rarely been worked on, i.e., southern leaf blight and Curvularia leaf spot. Promising accuracy has been achieved. The disease incidence identification accuracy for maize southern leaf blight and Curvularia leaf spot reached over 0.90 when the disease developed to middle to late stages (more than 20 days after pathogen inoculation).

#### 4.2.1. Multi-Source Features

In this study, we combined band reflectance and indices from UAV-MS and UAV-TIR to form five different feature sets for maize leaf spot disease incidence detection. The results showed that selecting different features in separate sampling dates (feature set II) led to higher accuracy than using the same features to establish a monitoring model (feature set III). The classification accuracy would further decrease if feature selection was not performed (feature sets I, IV, and V). This finding is consistent with the results of pine wood nematode monitoring using UAV-HS data [13].

MS indices that are found to be important for maize leaf spot disease monitoring mainly included two types of features, one related to leaf pigments (e.g., CI) and the other related to soil (e.g., OSAVI and $MTVI_1$) (Figure 9). Most selected spectral indexes in the model were pigment-related. When maize is infected with leaf spot disease, elliptical spots are generated on the leaves, which will destroy the pigments and cell structure of the leaves, resulting in distinguishable traits for leaf spot identification. In previous studies, the RE and NIR bands have been found to be sensitive to the incidence of maize diseases such as dwarf mosaic virus infection [60] and streak virus disease [20]. Our research found that the G band can also provide some information when monitoring maize leaf spot disease. Some soil-related indices have also been found to be important in this study probably because the leaf spot was similar to the soil color.

Two TIR features are used in this study (T and NDCT) and they are both selected in the classification model for all the four sampling dates (Figure 9). The TIR images reflect the canopy temperature of the vegetation. The canopy temperature of the diseased maize was higher than that of the healthy maize, because the pigment of the vegetation got damaged. To adjust the injury, the respiration significantly increased, resulting in the temperature rise [6]. However, due to the rain in Xinxiang on 1 September 2021, the temperature decreased; thus, the temperature difference in the canopy of diseased and healthy maize was not significant. This largely explains why the importance of TIR features decreased on September 2. It should be noted that only two TIR indicators were used in this study. More indicators related to TIR are to be considered in future research.

#### 4.2.2. Optimal Classifier for Maize Leaf Spot Disease Incidence Monitoring

Among the four maize leaf spot monitoring model algorithms (BPNN, RF, SVM, and XGBoost), RF performed the best regardless of the input features. This result is

consistent with a previous work monitoring incidence of wheat powdery mildew [17]. The outstanding performance of RF is probably attributed to its robustness as well as high tolerance for noise, outlier, and missing values. It is not easily disturbed by the limitations of a single decision tree and can effectively solve the problem of data imbalance. In contrast, BPNN is prone to falling into local minimization, resulting in a contradiction between their predictive and training abilities [48]. XGBoost has good generalization ability, fast learning speed, and approximation ability [51]. In this study, the model accuracy of XGBoost is slightly lower than RF. Considering the operational efficiency and prediction accuracy, XGBoost can also be utilized for the efficient monitoring of maize diseases.

*4.3. Early Monitoring of Maize Leaf Spot Disease Incidence*

The model accuracy for maize leaf spot disease incidence classification was the lowest on August 6 among the four sampling dates. The main reason is that the differences between healthy and diseased maize in the early stage were subtle. Additionally, the relatively low spatial resolution on the first two sampling dates may also have contributed to the low accuracy. As the damage to maize yield with severe diseases is irreversible, future research should invest more into early monitoring [61].

The feature selection results on August 6 showed that the indicators related to the RE band and pigment should be closely monitored in the early stages of maize leaf spot development. In addition, we also found that the B band was selected into the early monitoring model, which showed the potential of the B band in early monitoring of maize leaf spot disease [32,62]. Canopy temperature T and NDCT were also selected, indicating that temperature change was an important indicator for the early monitoring of maize leaf spot disease incidence.

Nevertheless, the accuracy of the early monitoring model was the lowest among the four sampling dates. One of the main reasons for the low accuracy may be attributed to that the healthy and diseased maize shared similar values of band reflectance (Figure 5) and vegetation indices (Figure 6) on August 6. This result indicated that the spectral information obtained with the MS sensor used in this study could not identify the susceptible maize in the early stage. It should be noted that this finding does not disqualify MS data for early monitoring of crop diseases. Instead, altering the band settings (e.g., center wavelength, band width) could be helpful [5,14]. Therefore, in future research, we suggest that researchers can first use HS data to find bands that are the most sensitive to the early stage of maize leaf spot, and further research on these bands to improve the efficiency and accuracy of its early monitoring. In addition, another reason why the early monitoring accuracy using UAV multi-source data is low is the mismatch between disease incidence location and the view angle of UAV sensors. When the maize leaf spot disease occurred, it was the bottom leaves that showed symptoms first, and then there was a gradual development to the upper leaves [63]. Hence, UAV could not fully obtain the spectral information of the susceptible maize in the early stage. Therefore, in the future, we should consider using oblique photography or ground photography to obtain near-ground data, to improve the accuracy of early disease monitoring.

**5. Conclusions**

This study proposed an approach for monitoring maize leaf spot disease incidence using UAV-based MS and TIR images and machine learning algorithms. The optimal model should use both spectral and thermal features, optimize the feature set with a feature selection algorithm, and employ a robust classifier. Among the four algorithms (BPNN, RF, SVM, and XGBoost) compared, RF was found to be the most accurate. Using features filtered with the RFE algorithm and the RF classifier, maize infected with leaf spot diseases can be successfully distinguished from healthy maize after 19 days of inoculation, with precision >0.9 and recall >0.95. Nevertheless, the accuracy was much lower (precision = 0.4, recall = 0.53) when disease development was in the early stages.

The canopy spectral response characteristics of healthy and diseased maize at different stages of inoculation were analyzed. In the visible light region, the canopy spectrum of maize leaf spot disease is higher than that of healthy maize, while in the near-infrared region, the spectral reflectance of the diseased canopy is lower. As the disease develops, the difference between healthy and diseased maize in the near-infrared region becomes increasingly apparent. In addition, the spectral reflectance of both diseased and healthy maize shows a trend of first increasing and then decreasing, which is related to the natural growth and development of maize. Most of the common features selected with the RFE algorithm are related to leaf pigments, while features related to TIR were selected at all four sampling dates with high importance.

In the future, more TIR features such as the crop water stress index should be further considered for monitoring diseases. To enhance the early disease monitoring, we can try to integrate HS data, fluorescence data, meteorological data, and other MS and TIR imaging data to establish a high-precision early monitoring model. UAV-based oblique photography should also be considered.

**Author Contributions:** Methodology, data curation, conceptualization, formal analysis, and writing (original draft preparation), X.J. (Xiao Jia); conceptualization, validation, formal analysis, and writing (review and editing), D.Y.; data curation, Y.B. (Yali Bai), X.Y., M.C., S.L., Y.B. (Yi Bai), L.M., Y.L., P.D., Q.L., F.N., C.N. and L.S.; software, Y.S.; funding acquisition, writing (review and editing), and conceptualization, X.J. (Xiuliang Jin) and W.G. All authors have read and agreed to the published version of the manuscript.

**Funding:** This research was supported by Central Public-interest Scientific Institution Basal Research Fund for Chinese Academy of Agricultural Sciences (CAAS-ZDRW202107, Y2022XK22), Nanfan special project, CAAS (ZDXM2310, YBXM01, YBXM20, YBXM2305), National Natural Science Foundation of China (42071426, 32271993), Research and application of key technologies of smart brain for farm decision-making platform (2021ZXJ05A03), The Henan Provincial Science and Technology Major Project (221100110100); The Joint Fund of Science and Technology Research Development program (Cultivation project of preponderant discipline) of Henan Province, China (222301420114).

**Data Availability Statement:** Data are available upon reasonable request; please email jinxiuliang@cass.cn if you need the data.

**Conflicts of Interest:** The authors declare no conflict of interest.

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
