# Peer review of "Monitoring Maize Leaf Spot Disease Using Multi-Source UAV Imagery"

_drones, doi:10.3390/drones7110650_

Round 1
Reviewer 1 Report
Comments and Suggestions for Authors
Jia et al. described leaf spot disease monitoring by Multi-sensor UAV imagery.
The manuscript is well written and for the first version in a good structure. However, serveral parts must be rewritten.
My major concerns are:
1. The disease severity in the plots is neither clearly descibed nor mapped and compared to sensor data.
2. Why was the sensor data acquied at different altitudes?
3. Most figures in the MS were not readable.
4. Very low number of ground truth data. Why did authors not do an extensive disease assessment?
5. Whytalking about feature and not bands and indices? Feature can be everything like extracted shapes of leaves or plants or soil. That is confusion. please rephrase throughout the manuscript.
6. In several graphics p-values / significance levels were indicated with *. Please double check them. There are several mistakes (Fig 6 & 7)
7. structure the results section by your findings, not by figures.
8. The discussion is poor. Authors are trying to interprete there data. However, they do not relate and discuss to findings of other research(e.g. 367-426 only four references)
Some more detailed comments:
Line 26/27: Temperature were all... ? Phrase without end.
32/33: ...third crop product in terms of what? ...second (what??) only to wheat and rice
58-67: all given references do not say a thing without giving disease severity and status.
66:Among these... What are "these"? there was no description of machine learning models.
67: Random Forest is a statistical method not a machine learning model.
70/75: Unclear. Scales mixed. Authors are talking about UAV RGB images (70) massive labeling (74) and leaf scale(75). Confusing. Rephrase.
75: "On the other hand" - there is missing "on the one hand..."
76: methodologically delete "ly"
76: "adopt" doesn't make sense
79: replace "of" by "for"
81-82: rephrase "Walter..."
98-101: delete
105: quantify "hot", "rainy", "humid"
108: there is no a and b in Fig 1
109: maize material = maize plants? Not clear what "materials" stands for.
109-110: "A total of 246..."; 226+8+17=251 double check and correct!
137-140: accuracy of temperature sensor?
Fig 2b: what is the small thing next to the MS sensor?
Tab 1: Aug 21 is 19 days after infection. Please correct.
145: "field data" means ground trouth data?
149-150: two plants per plot (how many plots) and three leaves per plant. that is a pretty low number. Why did you use such a low number? How can authors extrapolate to the whole plots from this low number.
Tab 2: Reference of the scale mising. Is the same scale used for both the diseases? Poor image quality.
169-172: delete
Fig 4: No link between visual and sensor data. How were these data sets combined and compared? That is the critical point in this manuscript. If this can't be clearly solved and explained, the manuscript is totally flawed.
179: "Second, and..." delete "and"
180: geographicalLY registratED
186: Where are the fround control points? Please indicate in Fig 1
190-191: give version of Jupyter Notebook and Python
Table 3: if only one line in one row then align the text to the middle of the row
202-205: unclear. Rephrase
220: delete "eXtreme GRadient Boosting and use only abbreviation like in the previous sections.
235-251: How did authors identify TP, TN, FS and FN? That sounds all good if you would have done the ground trouth in a sufficient manner. At least authors should have checked severeal plots completely and justify that 2 plants and 3 leaves per plot is enough!
256-258: delete sentence "Aug 6 was..."
Fig 6: How do authors know what is healthy and what is diseased? Make dots or bars but no line. There is no continuum between the bands. Hence it doesn't make sense to connect the dots with lines.
Fig 7: poor quality. What is the statistical test used? Why always three *** except for Blue? Aug 6 always significant?! Doesn't make sense
290-291: meaning of highest or lowest? Dosn't make sense! Rephrase.
Fig 8: see Fig 7
298: periods = sampling times?
299: TIR feature = TIR indices?
300: "local temperature changed trend"? rephrase and give tempreatur climate conditions over experiment period.
307: authors shold present their results by their findings and not start every section with "Figure x shows..." The figures help to understand the text. And all figures should be understandle on their own with their caption!
318: RFE?
321: here "feature" means something else again. Make sure not to mix different meaning into one word.
325: "...both original reflectance bands..." specify! What are the both bands? there 5 bands in the MS sensor and one band in the TIR sensor.
Fig 11: unclear
348: "...slightly higher or lower..." unlcear. Rephrase.
Fig 12: What ae the groups A-E? How were they established?
357: "...curve of healthy maize... early metaphase of infection..." How can the plants be healthy in a metaphase of infection? doesn't make any sense!
367 to end: discouss your findings to related research!
428: replace "than" by "to"
Comments on the Quality of English Language
Needs some coreection
Author Response
Dear Reviewer,
We are submitting here a revised manuscript entitled “Monitoring Maize Leaf Spot Disease Using Multi-source UAV Imagery” (drones-2629527). This manuscript was firstly submitted in Sep 2023. We are grateful to the three reviewers for their insightful suggestions and have revised the manuscript accordingly.
The following five major changes have been made are as follows:
- We have reorganized the structure of results section based on our findings and supported the statements with the figures.
- We have rewritten the discussion section. We compared our findings with previous research, confirmed some of their findings, and discussed the advantages and limitations of our proposed approach.
- We have rewritten the materials and methods section to better explain how the field data and UAV data are processed and linked.
- We have added quantitative information in the abstract and conclusions.
- We have remade the figures to make them easier to understand. Errors in the t test results are corrected.
One by one responses are provided.
Looking forward to your favorable decision.
Thank you.
With best regards,
Xiao Jia and co-authors

Reviewer 2 Report
Comments and Suggestions for Authors
The study of maize leaf spot disease using UAVs equipped with multispectral and thermal cameras and machine learning algorithms is a relevant topic. The recommendations to consider in the manuscript are listed below:
1. Summary: On the materials and methods, there is no mention of the use of vegetation indices as characteristics to be evaluated. On the results and conclusions, it should be deepened including numerical information on the results obtained and the most robust autonomous learning algorithm.
2. Lines 254-258 are repeated in the section on Materials and Methods, it should be considered whether to change the wording or delete.
3. The bibliography reference 24 is repeated in 47.
Author Response
Cover letter
Dear Editor and Reviewers,
We are submitting here a revised manuscript entitled “Monitoring Maize Leaf Spot Disease Using Multi-source UAV Imagery” (drones-2629527). This manuscript was firstly submitted in Sep 2023. We are grateful to the three reviewers for their insightful suggestions and have revised the manuscript accordingly.
The following five major changes have been made are as follows:
- We have reorganized the structure of results section based on our findings and supported the statements with the figures.
- We have rewritten the discussion section. We compared our findings with previous research, confirmed some of their findings, and discussed the advantages and limitations of our proposed approach.
- We have rewritten the materials and methods section to better explain how the field data and UAV data are processed and linked.
- We have added quantitative information in the abstract and conclusions.
- We have remade the figures to make them easier to understand. Errors in the t test results are corrected.
One by one responses are provided.
Looking forward to your favorable decision.
Thank you.
With best regards,
Xiao Jia and co-authors

Reviewer 3 Report
Comments and Suggestions for Authors
General comments
The study is crucial regarding disease monitoring in maize corp. The authors have conducted it well and in essential stages of disease occurrence. However, it needs some improvement regarding the article's presentation, the discussion part needs to be modified, and the importance of the machine learning classifiers in the introduction sections. Please give abbreviations of all acronyms. The manuscript needs to be modified in a major manner.
Specific comments:
Abstract: Name the classifiers and mention the prominent results in the abstract that readers can easily overview.
Line 17: Give the scientific name for the disease Leaf spot or the type of fungus that caused it.
Line 25-26: Revise sentence “while thermal features such as canopy temperature and normalized canopy temperature were all”. Please clear
Line 32: “crop product” is confusing. Please revise
Line 49: UAV MS replace with UAV-MS, UAV HS with UAV-HS, follow same order in whole manuscript.
Line 66 and 67: Irrelevant connection of sentence. The previous sentence focused on the type of data but suddenly changed to classification algorithms in the last sentence.
Line 71-72: Rewrite the sentence
Line 94: Objectives 2 and 3 do not show clear difference, seems intermixed
Figure 4. The workflow is not clear (Connect DI)
Figure 5, At which stage used the recursive feature elimination? Please also mention the disease studies used for RFE for disease monitoring. Several authors have used it.
Figure (f) 6, The usual behavior of the spectra under disease invasion is that at RGB bands, the reflectance increases and in NIR decreases, while in your case at the late stage it is decreasing. (Please recheck spectra results)
All the figures are not clear (Blurry), very difficult to read. Please make them clear.
Add references in the discussion part.
Comments on the Quality of English Language
The quality of English is moderate.
Author Response

(The authors gave the same response as above.)

Round 2
Reviewer 1 Report
Comments and Suggestions for Authors
The manuscript was well improved. However there are some issues, that could be solved:
Introduction: The introduction was clearly improved. However, please add the disease status of the cited examples. Authors added to the accuracy also the specificity and sensitivity. But I think the disease status e.g. progression of the disease, time after inoculation or disease onset, should be always given because a disease in a progressed state can be easily detected. However, practical information for the farmer needs to be of early disease symptoms or better symptoms that occur earlier than visible symptoms what can be realized with MS or HS data. See: https://www.frontiersin.org/article/10.3389/fpls.2019.00628
Fig4. I'd suggest to delete this figure because the same can be found in Tab2.
Fig5. This is a standard processing line and does not need any figure.
Fig6. is really good to get an overview!!
Discussion:
468-487: authors list research of other research groups. However, they do not really relate their results to the other findings. The section is more like part of the introduction. Please rewrite and discuss with your findings. And parts can be also mentioned in the introduction.
566-569: add the disease status to the precision; add space before the ">" letters
Author Response
Cover letter
Dear Editor and Reviewers,
We are submitting here a revised manuscript entitled “Monitoring Maize Leaf Spot Disease Using Multi-source UAV Imagery” (drones-2629527). This manuscript was firstly submitted in Sep 2023 and has gone through a round of major revision lately. We are grateful to the reviewer 1 for the further suggestions and have revised the manuscript accordingly.
The following three changes have been made are as follows:
- We have specified the disease status of crops for the cited publications and for our reported accuracy, in the introduction, discussion, conclusions, and abstract.
- We have more carefully stated the link between our research and previous studies, and moved general descriptions of the literature to the introduction section.
- We deleted redundant figures (Figure 4 and Figure 5).
One by one responses are provided.
Looking forward to your favorable decision.
Thank you.
With best regards,
Xiao Jia and co-authors

Reviewer 3 Report
Comments and Suggestions for Authors
The authors have revised carefully
Author Response

(The authors gave the same response as above.)
